# Rational Use of Antibiotics in Neonates: Still in Search of Tailored Tools

**DOI:** 10.3390/healthcare7010028

**Published:** 2019-02-16

**Authors:** John van den Anker, Karel Allegaert

**Affiliations:** 1Division of Clinical Pharmacology, Department of Pediatrics, Children’s National Health System, 111 Michigan Avenue NW, Washington, DC 20010, USA; JvandenA@childrensnational.org; 2Division of Paediatric Pharmacology and Pharmacometrics, University of Basel Children’s Hospital, Spitalstrasse 33, 4056 Basel, Switzerland; 3Intensive Care, Erasmus MC-Sophia Children’s Hospital, Dr Molewaterplein 40, 3015 GD Rotterdam, The Netherlands; 4Department of Pediatrics, Division of Neonatology, Erasmus MC-Sophia Children’s Hospital, Dr Molewaterplein 40, 3015 GD Rotterdam, The Netherlands; 5Department of Development and Regeneration, KU Leuven, Herestraat 49, 3000 Leuven, Belgium

**Keywords:** newborn, rational drug use, antibiotics, safety, stewardship

## Abstract

Rational medicine use in neonates implies the prescription and administration of age-appropriate drug formulations, selecting the most efficacious and safe dose, all based on accurate information on the drug and its indications in neonates. This review illustrates that important uncertainties still exist concerning the different aspects (*when, what, how*) of rational antibiotic use in neonates. Decisions *when* to prescribe antibiotics are still not based on robust decision tools. Choices (*what*) on empiric antibiotic regimens should depend on the anticipated pathogens, and the available information on the efficacy and safety of these drugs. Major progress has been made on *how* (beta-lactam antibiotics, aminoglycosides, vancomycin, route and duration) to dose. Progress to improve rational antibiotic use necessitates further understanding of neonatal pharmacology (short- and long-term safety, pharmacokinetics, duration and route) and the use of tailored tools and smarter practices (biomarkers, screening for colonization, and advanced therapeutic drug monitoring techniques). Implementation strategies should not only facilitate access to knowledge and guidelines, but should also consider the most effective strategies (‘skills’) and psychosocial aspects involved in the prescription process: we should be aware that both the decision not to prescribe as well as the decision to prescribe antibiotics is associated with risks and benefits.

## 1. Introduction: Why a Review on the Rational Use of Antibiotics in Neonates is Valuable

The rational use of medicines is defined by the World Health Organization (WHO) with the intention to assure that every single patient receives the medicines appropriate for his/her needs, with doses in accordance with his/her individual characteristics, for the appropriate duration and—preferably—at a reasonable cost [1]. The commonly used indicators of potential irrational practices are the incidence of polypharmacy, inappropriate self-medication, failure to adhere to clinical guidelines or dosing regimens, or the inappropriate use of antibiotics (dose, indication, duration, route of administration) [1]. The application of these clear and obvious definitions to rational antibiotic use in neonates has major limitations [2,3]. These limitations relate to the extensive variability in clinical practice across the globe (e.g., empiric prescription of antibiotics, extensive variability in dosing practices, large differences in body weight of neonates) in combination with the limited evidence to support any of these practices. As a consequence of this extensive variability and the limited evidence, the commonly applied research tools to audit the rational use of antibiotics fail when applied to the neonatal population [3,4,5].

To further illustrate this, these tools include the concept of defined daily dose (DDD, average maintenance dose when used to treat its major indication in adults). Although Liem et al. reported on an effort to develop neonatal DDD based on an assumed neonatal weight of 2 kg, the same researchers very recently had to conclude that standardized average daily dosages of commonly used antibiotics were not achieved [3,4]. Similarly, Laine et al. also recently reported on an attempt to quantify the extent of off-label use of antimicrobials in neonates, but also concluded that the DDD tool has major limitations when applied to neonates, and that novel, more accurate methods to evaluate antibiotic use in neonates are urgently needed [5]. 

Other research method-related aspects that are of relevance for antibiotic consumption studies in neonates relate to the study design, such as the statistical descriptive analysis used [6]. To further illustrate the latest issue raised, we refer to the most recently reported study on medicine use by Pediatrix, the leading provider of maternal–fetal and neonatal care in the United States. In its analysis (2005–2010, n = 450,386 neonates) of medicine use, polypharmacy was common with a mean number of medicine courses of 4 (1–14) per newborn, and 17 (2–45) courses in extremely low birth weight (<1000 g, ELBW) neonates. More relevant to the topic of this paper, 26 different antibiotics were classified in the top 100, 16 in the top 50, and three (ampicillin, gentamicin, vancomycin) were in the top five of the most commonly administered drugs (Table 1) [7]. However, reflecting on the impact of the analytical approach taken, the final sequence also depends, in part, on how the data were analyzed: either based on exposure (number of neonates exposed during admission/1000), or on the number of courses (number of courses/1000 admitted neonates), or on the days of use (days of use/1000 days admitted on the unit). 

Along the same line on the relevance of research tools to assess the use of medicines in neonates, Nellis et al. [8] compared two cross-sectional study designs (a service evaluation survey for 3 days, compared to a single day web-based point prevalence study). The service evaluation survey resulted in a significant higher likelihood to capture a given compound. Unfortunately, this study did not report on observations specific to antibiotics [8,9]. However, the same European Study of Neonatal Exposure to Excipients (ESNEE) research group also reported on the prescription practices of antibiotics in neonates in a point prevalence study. In this dataset, the 12 most commonly prescribed antibiotics (in descending order) were gentamicin, penicillin G, ampicillin, vancomycin, amikacin, cefotaxime, ceftazidime, meropenem, amoxicillin, metronidazole, teicoplanin and flucloxacillin, covering 92% of all antibiotics prescribed. This sequence is rather similar to the sequence described by Hsieh in the US cohort (Table 1) [7,8,9]. Finally, another point prevalence study on antibiotic prescribing included 589 neonates in 50 neonatal units from 14 different European countries (Antibiotic Resistance and Prescribing in European Children, ARPEC). This study included 1712 neonates, of whom 532 (31%) were exposed to antibiotics in the point prevalence study, ranging from 19.9% in a general neonatal ward to 39% in a neonatal intensive care setting [10]. 

Based on the illustrations provided, it should at least be obvious that exposure to antibiotics is very common as these medicines are the most frequently prescribed (Anatomical Therapeutic Chemical Classification, ATC) group in hospitalized neonates. Obviously, frequent prescription does not in itself reflect rational medicine use. This is because rational medicine use implies the prescription and administration of a safe formulation using the optimal dose, both based on accurate information on the medicine and its indications in neonates [11,12]. In this review, we will illustrate that important uncertainties still exist on the different aspects (when, what, and how, most prominent on the ‘when’ aspect) of antibiotic use in neonates, all related to rational drug use. To make relevant progress on the rational use of antibiotics in neonates, additional tools validated in this population are needed.

## 2. When to Prescribe Antibiotics in Neonates

At present, we still lack sufficiently robust decision tools to disentangle (sensitivity vs. specificity) infectious events from other commonly observed events in neonates. This is likely even more relevant for early onset sepsis (EOS). Up to 10% of newborns are screened, while only a minority of these screened cases (<5%) are subsequently classified as having EOS (either culture proven, or based on clinical assessment or biomarker driven ‘blood-culture negative sepsis’) [13,14]. This results in an incidence of culture-confirmed EOS of about 0.4–0.8/1000 in term neonates, and about a 10-fold higher incidence of cases classified as culture-negative sepsis [14]. This over-prescription can be explained by the relevant mortality (10–12%) and morbidity related to this diagnosis.

Prediction tools have been developed, but do not yet have sufficient sensitivity and specificity to be accepted by the community. The use of an EOS calculator (gestational age, maternal antepartum temperature, duration of rupture of membrane, maternal Group B Streptococcus status, intrapartum antibiotics, clinical exam = well appearing, equivocal, clinical illness; threshold for empirical prescription of antibiotics = risk > 3/1000) resulted in a decrease in blood culture use and empirical antibiotic administration from 14.5 to 4.9% and 5.0 to 2.6%, respectively, without apparent negative effects [15]. For late onset sepsis (LOS), prediction tools for bloodstream infection based on deviant heart rate characteristics (HRC score, driven by decreased variability and transient decelerations) have been developed, although these tools also have their limitations [16]. As reported by Coggins et al., elevated HRC scores had limited ability to detect blood stream infections (BSI), since these scores were often elevated in NICU patients who had no bloodstream infection. In infants with blood stream infections (n = 46), 37% had at least one HRC score >2, and 11% had at least one HRC score >5, suggesting that both the sensitivity and specificity of the current tools are still poor [17].

In this specific population, false negative blood cultures can be explained by maternal and subsequent fetal exposure to antibiotics, or more commonly, because the volumes of blood collected were too small [14,18]. In a recent German survey, the majority (59%) of respondents aimed to collect 0.5 mL. In a laboratory setting and using a quantitative approach, it has been suggested that 1.0 mL has excellent sensitivity even in the setting of quantitative low bacteremia, while 0.5 mL is sufficient to detect moderate or high grade bacteremia [19]. 

Due to these limitations in the sensitivity and specificity of the biomarkers currently available in the routine clinical setting, structured guidelines may result in more consistent practices, but also, prolonged antibiotic exposure (+10 h), an increase in the number of lumbar punctures (14 to 23%) and delayed discharge (length of stay >5 days: 20.9 to 27.7%) in term newborns, as has been reported following the implementation of the CG149 NICE guidelines in the United Kingdom [20]. In contrast, procalcitonin (NeoPins study) turned out to be superior to standard care in reducing antibiotic therapy in neonates with suspected early-onset sepsis, with a median reduction in the duration of exposure from 65 to 55 h. Non-inferiority for re-infection or death could not be shown due to the low occurrence of re-infections and absence of study-related death [21]. 

## 3. What Antibiotic Regimen to Prescribe in Neonates

The choices to be made when prescribing empiric antibiotic regimens should depend on the anticipated pathogens and their resistance pattern, and on the available information on the safety and side effects of the antibiotics, specifically in neonates. Obviously, once the pathogen is isolated, the best targeted and tailored treatment regimen should be selected [22]. Pathogens to target for in EOS are either Gram-negative organisms (*Escherichia coli*, *Haemophilus influenzae*, *Citrobacter* spp., or *Enterobacter* spp.) or Gram-positive organisms (*Streptococcus agalacticae*, *Streptococcus viridans*, *Listeria monocytogenes*, *Staphylococcus aureus* or *Pneumococcus*). LOS is most commonly due to coagulase-negative Staphylococcus, as well as Gram-negative organisms, and *Staphylococcus aureus* can be isolated. It is hereby important to be aware that both EOS and LOS are associated with relevant morbidity (including neurocognitive outcome) and mortality, and this is also the case of coagulase-negative staphyloccal infections [23,24].

For EOS, a very recently published survey on 80 German NICUs reconfirmed that the majority (89%) of units prescribe a combination of a beta-lactam (mainly ampicillin, sometimes penicillin G) and an aminoglycoside (gentamicin > tobramycin > amikacin), while only a minority use piperacillin (11%), piperacillin-tazobactam (4%), cefotaxime (10%) or ampicillin-sulbactam. For LOS, practices were much more heterogeneous, with third-generation cephalosporins (52%), or carbapenems (meropenem 18%), while 48% of the units use vancomycin empirically before any methicillin-resistant Gram-positive pathogen is detected [18]. These practices are, to a large extent, in line with reported practices in other countries or regions [7,13,22]. However, drug choices are relevant when we consider potential side effects. 

In our assessment, routine TDM for aminoglycosides in the first day(s) of EOS empiric treatment to avoid oto- or nephrotoxicity is not necessary, unless in very specific settings (asphyxia, extra-corporeal membrane oxygenation, congenital renal dysfunction), but TDM is very relevant when aminoglycoside exposure is continued beyond 48–72 h [25]. In contrast, vancomycin administration warrants TDM because there is still uncertainty on the dosing regimens (cf. infra, 4.). Although the use of broad spectrum antibiotics may be perceived to be more convenient since this avoids these potential toxic effects and the need for TDM, the use of third-generation cephalosporins or carbapenems is associated with a clinically relevant higher risk to developing invasive fungal infections [26]. 

Moreover, it has been well documented that the antibiotic policy prevents the emergence of resistant bacilli in neonates. When comparing two commonly applied antibiotic policies for empiric treatment (cefotaxime + amoxicillin vs. penicillin G + tobramycin), an 18-fold higher risk to observed colonization with resistant pathogens was documented in favor of the penicillin G + tobramycin regimen [27]. 

## 4. How to Prescribe Antibiotics in Neonates

It is reasonable to assume similarity in antimicrobial pharmacodynamics, so that similar time–concentration profiles should be aimed for in neonates when compared to other populations [28,29]. Three key patterns have been defined, depending on the mechanisms of the specific antibiotic to attain maximal bacterial killing. These patterns are either (i) the proportion of time that a given antibiotic remains above a given minimal inhibitory concentration (MIC) value (such as beta-lactams); (ii) the peak concentration of a given antibiotic above a given value (such as aminoglycosides); or (iii) a mix of both, with the area under the concentration–time curve above a given target concentration (antibiotic specific)(such as vancomycin) [29,30]. These patterns are not different in neonates, so the main effort relates to tailoring the doses administered to the PK characteristics of the newborn [30]. 

Intravenous drug administration is hereby the most commonly used route of administration in hospitalized newborns, although there are also challenges (such as slow intravenous flow rates, small drug volumes, effects related to the dead space or flush volumes) related to this route that may result in delays or variability in the rate of drug delivery in neonates [31]. Likely more relevant, considerable between-unit differences in antibiotic dosing regimens were observed both in a survey in 44 French units, as well as in a European point prevalence study on antibiotic dosing in the ESNEE study [32,33]. The development of online (inter)national pediatric drug formularies, at best disconnected from the different clinical syndrome-driven guidelines (such as EOS, LOS, NEC), was effective in the Netherlands. Such an approach is especially of relevance in the setting of a lack of evidence-based dosing guidelines, and applied a framework to provide dosing guidelines based on the best available evidence from registration data, investigator-initiated research, professional guidelines, clinical experience or consensus [34]. Once there is reasonable consensus, neonatal DDD can be developed. 

### 4.1. Beta-Lactam Antibiotics

The bactericidal effect of beta-lactams is by interfering with peptidoglycan cross-linking and subsequent interfering with the bacterial cell wall structure. The distribution of antibiotics is driven by maturational differences in body composition (water%). Protein binding may also be of relevance for some of the protein-bound beta-lactam antibiotics, such as cefazolin or ceftriaxone, since it is the fraction of time (%fT) during which the free antibiotic is above a given MIC (% *f*T > MIC is the target). Total plasma protein increases in early infancy to reach adult levels at 10 to 12 months. Within the neonatal age range, albumin significantly increases with postmenstrual age, but this only explained 20% of the variability, suggesting that there are also non-maturational factors involved [35,36]. These maturational differences affect the ratio between the total and the unbound (active) concentrations, as modeled for cefazolin in neonates [37]. 

Subsequent elimination is almost exclusively by renal elimination, but covers both glomerular filtration as well as renal tubular transport (excretion and absorption). The main factors involved in the development of renal function are gestational age (GA) and the dramatic sequential hemodynamic changes after birth following a fetal setting dominated by high vascular resistance and lower renal blood flow. The postnatal increase is due to the increased cardiac output and the reduced renal vascular resistance, resulting in increased renal blood flow, changes in intra-renal blood flow distribution, and higher permeability of the glomerular membrane [38,39]. This translates in dosing guidelines that consider gestational age or birth weight and postnatal age, as reflected in the dosing regimens for penicillin G (25,000 IU/dose, q6h–8h–12h based on birth weight/gestational and postnatal age) [40], cefazolin (weight and postnatal age) [37], cefotaxime (gestational and postnatal age) [41] and meropenem (20–40 mg/kg, q8h depending on gestational age and MIC) [42]. Interestingly, this last paper also suggested an evaluation of the concept of prolonged infusion. Using this approach, the same 24 h dose exposure will result in a higher time above a given MIC (Figure 1). Recently, this approach (20 or 40 mg/kg, but over 4 h instead of 30 min) has been proven to result in improved survival and faster reduction in inflammation markers in a cohort of 102 neonates with proven Gram-negative sepsis [43]. 

### 4.2. Aminoglycosides

The dosing regimens for aminoglycosides should maximize the C_max_ (peak/MIC > 8–10) to attain efficacy, while lower trough values (C_trough_) should be aimed for to avoid toxicity (renal, hearing). This has been translated in the ’extended daily dosing interval’ concept. 

Since aminoglycosides are water soluble (distribution, C_max_) and are subsequently cleared by glomerular filtration (renal, C_trough_), this results in a CATCH-22 situation in neonates. Due to the higher relative distribution volume in neonates (L/kg, body water content), a higher mg/kg dose is needed, while the reduced glomerular filtration subsequently necessitates a further extended time interval, sometimes beyond 48 h as validated for an amikacin dosing regimen in neonates, with an additional extension of 12 h (*****) in the setting of asphyxia and hypothermia, and 10 h (******) when ibuprofen is co-administered (Table 2) [44,45,46]. A similar approach with higher doses/administration and further extended dosing intervals has been suggested for gentamicin and tobramycin, with additional adaptations in the setting of asphyxia and hypothermia, or when ibuprofen is co-administered [47,48,49].

### 4.3. Vancomycin

The area under the total concentration–time curve (0–24 h, AUC_24h_) divided by the MIC (pathogen-specific) (AUC_24h_/MIC) of 400 is the most commonly accepted efficacy target to guide vancomycin dosing, with an upper limit of 700–720 to avoid an unacceptable toxicity risk (mainly renal). However, this target is based on *Staphylocccus aureus* pneumonia data in adults, not really reflecting neonatal clinical infections [25,50]. Current consensus guidelines recommend measuring trough vancomycin concentrations during intermittent dosing as a surrogate for the AUC_24h_ [25,50]. 

One approach to improve this setting is use of the personalized Bayesian dose adjustments tools of vancomycin tailored to neonates, such as the DosOpt [51]. As illustrated by Zhao et al., this necessitates further external validation since the external predictive performance validation of six published vancomycin models in neonates underperformed. This underperformance was only in part explained by differences in the analytical techniques (assays) for creatinine and vancomycin [52]. Maturational differences in vancomycin protein binding are likely also of relevance, since the free vancomycin fraction is higher in neonates when compared to children or adults [53]. 

### 4.4. Other Routes of Administration

While the higher discussed intravenous drug administration is the most commonly used route of administration in hospitalized newborns, there are some other routes (intramuscular, oral) worth considering. 

Intramuscular administration turned out to be effective in the African Neonatal Sepsis Trial (AFRINEST) studies in neonates with suspected infection in poor resource and outpatient settings [54]. A relevant physiological factor that influences drug absorption from an intramuscular injection site is the blood flow to and from the injection site and the muscle mass (total surface area). The muscle activity also displays maturational (age) and non-maturational (critical illness, neuromuscular diseases, muscular relaxants) covariates [30]. *Oral* antibiotics lead to slower absorption, lower bioavailability and more variability in neonates compared to parenteral administration, but adequate serum levels can be reached. The antibiotic duration and timing of the switch from an intravenous to oral route for bacterial infections in children have recently been reviewed, but similar concepts need further exploration in neonates [55]. 

### 4.5. Duration of Treatment

When considering the duration of treatment for EOS or LOS and its efficacy, there is only very limited evidence on the duration, except for the fact that the pathogen is relevant [55]. We aim to provide some pieces of information on the blood culture positive clinical outcome data in neonates to further illustrate the complexity. Two studies assessed the outcome in a variety of isolated pathogens [56,57], and two studies assessed pathogen-specific outcome [58,59]. 

Gathwala et al. randomized 60/181 (pre)term neonates (>1.5 kg) with a positive blood culture (72% EOS, order of frequency: *Pseudomonas*, *Acinetobacter*, *Klebsiella*, *Enterobacter*, *Escherichia coli*, *Staphylococcus aureus*) that were recovered clinically on day 7, to either 10 or 14 days. In this setting, there were no significant differences between both groups, except for a shorter hospitalization (mean 13 vs. 17.5 days) in the 10 days group [56]. Rohatgi et al. randomized 132 (pre)term neonates (>1.5 kg) blood culture (Staphylococcus epidermidis was an exclusion criterium) positive results (65% EOS, in order of frequency: *Klebsiella* spp., *Staphylococcus Aureus*, *Enterobacter*, *Methicillin Resistant Staphylococcus Aureus*, *Enterococcus*, *Acinetobacter*) in two different groups (duration of treatment) [57]. There was a shorter hospitalization in the 7 days group (mean 17 versus 19.4 days) when compared to the 10 days group, without any other differences in outcome, including safety aspects. Both studies were conducted in a single Indian neonatal intensive care unit [56,57].

Linder et al. reported on a retrospective analysis of medical files of 126 very low birth weight infants with late onset coagulase-negative Staphylococcus sepsis, treated for 5, 6–7, 8–10, or >10 days in 38%, 25%, 24% and 12% of the sample, respectively, after the last positive blood culture. Based on their analysis on the safety and outcome, these authors concluded that treatment with vancomycin for 5 days after the last positive blood culture was associated with a satisfactory outcome without adverse effects [58]. In a small trial in 69 neonates with a positive blood culture, Chowdhary et al. observed that treatment failure in cases randomized to 7 days was more common when compared to 14 days (4/7 vs. 0/7 cases) in *Staphylococcus aureus*-positive cases [59].

The duration of treatment is not only related to a potential risk of efficacy failure (when too short), but is also of relevance when considering safety issues (when too long). A recent meta-analysis confirmed that prolonged antibiotic exposure in uninfected preterm neonates was associated with a higher risk for necrotizing enterocolitis or death [26]. These findings become even more relevant when we consider the culture-negative neonatal sepsis concept, in need of balanced decisions on efficient sepsis care as well as on antimicrobial stewardship [14].

## 5. Discussion: How to Improve the Current Setting

Rational antibiotic use implies the prescription and administration of a safe formulation using the optimal dose, based on accurate information on the medicine and based on a valid indication in neonates [11,12]. Based on the issues discussed on when, what and how, it is obvious that this is not yet the case for antibiotic use in neonates. This discussion provides a framework on how progress can be made to further ensure rational (safe and effective) antibiotic use in neonates.

### 5.1. Understand Neonatal Pharmacology

There is a significant improvement in the knowledge on pharmacokinetics in neonates, and the impact of the maturational and non-maturational covariates on dosing regimens (how to achieve the target exposure. This has resulted in an increased volume of scientific knowledge, but we still fail to implement this knowledge in our daily practice [14,18,22]. In contrast, the knowledge on the short- and mainly long-term side effects of perinatal exposure to antibiotics (pharmacodynamics) is still much more limited and is an area in need of further research. This also includes aspects such as the risk of developing atopy or obesity [26,60]. This research should include the search for mechanisms, since the long-term effects are at least in part explained by their (even transient) impact on the intestinal microbiome [60].

### 5.2. Let Us Make Our Practices More Targeted and Smarter

At the different levels (when, what and how), we can make more targeted decisions. Certainly, there is a lot of progress to be made in the prediction models and better use of the early biomarkers of sepsis, such as IL-6 or pro-calcitonin. Similarly, the use of structured screening for colonization may assist us to improve our knowledge on the pathogens to target when we have to make a choice on empiric-unit or patient-specific antibiotic regimens [18]. The integration of more advanced TDM techniques and Bayesian prediction tools [25,51] may hereby assist us to reduce the burden and improve the decisions made based on individual TDM observations. However, such tools need prospective validation in the population of interest. 

### 5.3. We Should be Aware of Our Limitations

As already mentioned in the Special Issue, we like to believe that decisions on medicine use in neonates are driven by rational processes, but we should also explore the psychosocial aspects that guide our decisions [11,61]. Progress on the rational use of antibiotics does not only depend on the availability of high quality data on efficacy and safety (‘knowledge’) and the related new tools (‘methods’), but also on the subsequent approaches taken to streamline access to such data and the development of implementation strategies. Data access and implementation strategies should not only facilitate technical access to data or guidelines, but should also consider the most effective strategies (‘skills’) to approach caregivers not only as rational decision makers, but also cover the psychosocial aspects involved in the decision process of the prescription of medicines [11,61]. The recent German survey hereby illustrated again the significant gap between the available guidelines and the daily practices in the different units [18]. We should at least be aware that both the decision not to prescribe antibiotics, as well as the decision to prescribe or continue antibiotics, will result in potential risks and benefits. 

## Figures and Tables

**Figure 1 healthcare-07-00028-f001:**
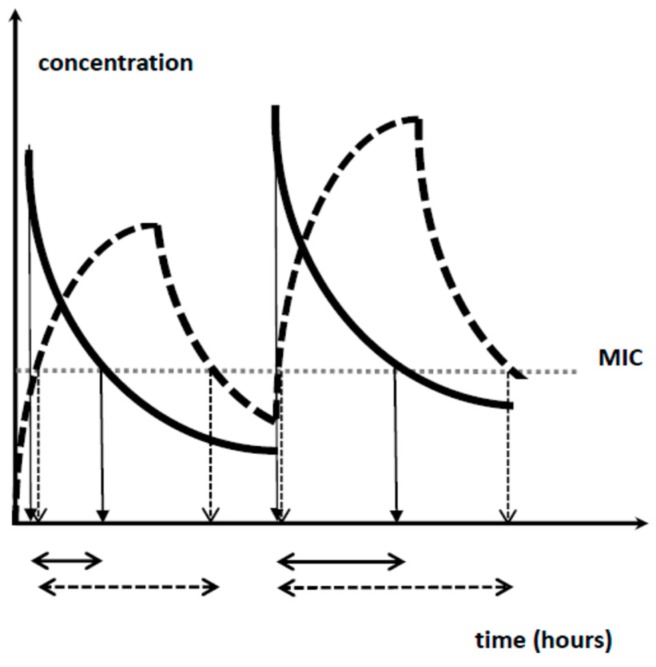
Illustration on how prolonged infusion of the same dose results in a higher fraction of time above a given minimal inhibitory concentration (MIC) value, as applicable for beta-lactam antibiotics.

**Table 1 healthcare-07-00028-t001:** The list of the 100 most commonly prescribed medicines in the neonatal intensive care unit, as published by Hiesh et al., contains 26 different antibiotics [7].

Rank	Medicine	Exposure	Courses	Days of use
**1**	ampicillin	681	709	3069
**2**	gentamicin	676	785	3521
**5**	vancomycin	91	150	987
**15**	cefotaxime	43	53	316
**23**	tobramycin	24	34	189
**24**	erythromycin	24	25	103
**28**	clindamycin	17	19	128
**38**	ceftazidime	12	15	99
**41**	piperacillin/tazobactam	11	15	115
**43**	amoxicillin	11	12	72
**44**	metronidazole	11	13	97
**45**	oxacillin	10	13	97
**46**	nafcillin	9	14	97
**48**	amikacin	8.8	12	77
**51**	cefazolin	7.5	8.1	27
**52**	meropenem	7	8.9	82
**60**	cefipime	6.1	7.7	58
**66**	penicillin G	4.7	4.9	38
**74**	rifampin	3.6	3.8	36
**77**	imipenem + cilastatin	3.0	3.3	29
**90**	cephalexin	1.9	2.0	9.5
**91**	ceftriaxone	1.8	1.8	5.7
**94**	sulfamethoxazole + trimethoprim	1.6	1.8	16
**96**	cefoxitin	1.5	1.6	4.9
**98**	fosphenytoin	1.4	1.6	10
**100**	linezolid	1.3	1.6	14

**Table 2 healthcare-07-00028-t002:** Amikacin dosing regimen following prospective validation [44,45,46].

Weight (gram)	Postnatal Age < 14 Days *^,^**	Postnatal Age ≥ 14 Days
<800 g	16 mg/kg/48 h	20 mg/kg/42 h
800–1200 g	16 mg/kg/42 h	20 mg/kg/36 h
1200–2000 g	15 mg/kg/36 h	18 mg/kg/30 h
2000–2800 g	15 mg/kg/36 h	18 mg/kg/24 h
≥2800 g	15 mg/kg/30 h	18 mg/kg/20 h

(*): +12 h in the setting of hypothermia + asphyxia; (**): + 10 h when ibuprofen is co-administered.

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
