# Peer review of "Rational Use of Antibiotics in Neonates: Still in Search of Tailored Tools"

_healthcare, 2019, doi:10.3390/healthcare7010028_

Reviewer 1 Report

General statement

I apologize for the slight delay in reviewing this paper due to medical problems.

This review is both interesting and somewhat disappointing.  Its interest stems from the fact that, as pointed out by the authors, there a few extensive reviews on the appropriate use of antibiotics in neonates (for a number of reasons).  Thus, filling up this gap is needed and may interest clinicians and other health care professionals dealing with neonates.  The authors must be commended for having taken this task.

However, it is disappointing since the authors come with very few additional suggestions and guidance than what has been already proposed in the past. 

Thus, for instance, the section on beta-lactams is very “classical” and we do not learn much more than what has already been extensively published for adults except that prolonged infusion has been successfully applied to neonates.  What about more recent progresses such as continuous infusion on the one hand and systematic monitoring of blood levels on the other hand?  For aminoglycosides, most of the recommendations are well known.  For vancomycin, the authors cold have mentioned that continuous infusion has become very popular (at least in Europe) and that even US-based investigators are now moving to true AUC/MIC-based optimization (rather than using only Cmin as surrogate; see Finch et al. A Quasi-Experiment To Study the Impact of Vancomycin Area under the Concentration-Time Curve-Guided Dosing on Vancomycin-Associated Nephrotoxicity. Antimicrob Agents Chemother. 2017 Nov 22;61(12). pii: e01293-17). Lastly, what about linezolid and its monitoring (to avoid thrombocytopenia ?.

The review also shows its limitations when discussing which antibiotic should be used.  The authors mention that there is a huge variability between centers (and reports) especially for late onset sepsis.  But this is not a surprise if taking into consideration that the choice of an antibiotic is primarily guided by the local epidemiology and the most likely local pattern of resistance.  These two aspects may vary very much between centres and between the dates at which the patients were treated (which can be years before the actual publication).  One may even ask whether it is really valid to use data from one center to apply to another one without assessing first that the epidemiology and the resistance patterns are similar.

Lastly, it would have been interesting if the authors would have come with more definite suggestions as how to improve the current settings.  What they propose is very general.  What about more specific changes that could impact the daily practice such as (i) rapid diagnostic approaches (including not only the identification of the potentially causative organism(s) but also of its/their high susceptibility or resistance to locally available antibiotics), or (ii) the use of biomarkers to help the clinician to stop antibiotic use (such as procalcitonin in adults), (iii) or the systematic monitoring of antibiotic blood levels as a way to correct for difficulties in dosage due to the important and often unanticipated variations in drug distribution and elimination.  I understand that many recommendations along those lines would be still far from being usable in daily practice but prospective views (with critical appraisal) would make the review (much) more interesting.  Incidentally, the first part of the paper seems to be very concerned with difficulties in assessing and measuring antibiotic use in neonates (we all know that adult DDDs cannot be used), but the authors make no real suggestion here.  Why not ?

Author Response

This review is both interesting and somewhat disappointing.  Its interest stems from the fact that, as pointed out by the authors, there a few extensive reviews on the appropriate use of antibiotics in neonates (for a number of reasons).  Thus, filling up this gap is needed and may interest clinicians and other health care professionals dealing with neonates.  The authors must be commended for having taken this task. However, it is disappointing since the authors come with very few additional suggestions and guidance than what has been already proposed in the past. We agree to a large extent on the frustration that rational antibiotic use in neonates is obviously needed, but difficult to describe since we still lack the tools to be more effective. In line with the replies to the other reviewers, we have tried to further focus on this, by adding additional stress on these tools, including also changing the title, and illustrations on how interventions can be both effective or not (NeoPins vs NICE guideline), or on how dosing regimens can be harmonized (national formulary) in a setting of off label practices in need of structured analysis of the available data to improve practices Thus, for instance, the section on beta-lactams is very “classical” and we do not learn much more than what has already been extensively published for adults except that prolonged infusion has been successfully applied to neonates. As both clinical neonatologists, we can only state that such practices are much more limited in neonates, in part also because of the logistics (access and number of accesses), so these successes (survival !) are important to share.   What about more recent progresses such as continuous infusion on the one hand and systematic monitoring of blood levels on the other hand?  For aminoglycosides, most of the recommendations are well known.   Although claimed to be more effective, we are not aware of any study that really documented superiority of eg vancomycin continuous administration with again potential risks related to accesses needed and resistance. We have received reported on these aspects (Pauwels et al, Arch Dis Child ; Allegaert 2018, Acta Clin Belg), and took the decision not the repeat this to avoid perceived issues related to plagiarism. For vancomycin, the authors could have mentioned that continuous infusion has become very popular (at least in Europe) and that even US-based investigators are now moving to true AUC/MIC-based optimization (rather than using only Cmin as surrogate; see Finch et al. A Quasi-Experiment To Study the Impact of Vancomycin Area under the Concentration-Time Curve-Guided Dosing on Vancomycin-Associated Nephrotoxicity. Antimicrob Agents Chemother. 2017 Nov 22;61(12). pii: e01293-17). Lastly, what about linezolid and its monitoring (to avoid thrombocytopenia ? Similar and although these comments are well taken, we are not aware of any large report on its established use in neonates, while the most recent reports on dosing practices in neonates also reflect the very limited use of this approach (Metsvaht 2015; Leroux 2015). To further illustrate the need to tailor such tools to neonates, nephrotoxicity in neonates is not a very easy task, since commonly used biomarkers like creatinine change with age, weight, postnatal age, and indeed clinical characteristics< The review also shows its limitations when discussing which antibiotic should be used.  The authors mention that there is a huge variability between centers (and reports) especially for late onset sepsis.  But this is not a surprise if taking into consideration that the choice of an antibiotic is primarily guided by the local epidemiology and the most likely local pattern of resistance.  These two aspects may vary very much between centres and between the dates at which the patients were treated (which can be years before the actual publication).  One may even ask whether it is really valid to use data from one center to apply to another one without assessing first that the epidemiology and the resistance patterns are similar. We have added data on the late onset sepsis and the pathogens involved, but also on how dosing regimens also affect resistance patterns within given units. Lastly, it would have been interesting if the authors would have come with more definite suggestions as how to improve the current settings.  What they propose is very general.  What about more specific changes that could impact the daily practice such as (i) rapid diagnostic approaches (including not only the identification of the potentially causative organism(s) but also of its/their high susceptibility or resistance to locally available antibiotics), or (ii) the use of biomarkers to help the clinician to stop antibiotic use (such as procalcitonin in adults), (iii) or the systematic monitoring of antibiotic blood levels as a way to correct for difficulties in dosage due to the important and often unanticipated variations in drug distribution and elimination.  I understand that many recommendations along those lines would be still far from being usable in daily practice but prospective views (with critical appraisal) would make the review (much) more interesting.  Incidentally, the first part of the paper seems to be very concerned with difficulties in assessing and measuring antibiotic use in neonates (we all know that adult DDDs cannot be used), but the authors make no real suggestion here.  Why not ? We agree, and have shifted to overall message of the paper to the tools to focus on that need improvement before progress can be made. The procalcitonin use has been mentioned in the revised version of the paper (NeoPins, added). TDM monitoring also necessitates the validation of predicting tools in neonates to be confident that adaptations are indeed on target (added). Neonatal DDD can only be developed once there is reasonable consensus on the dosing based on the clinical characteristics of the individual newborn (added).

Reviewer 2 Report

The rational use of antibiotics is an important public health issue that needs to be discussed for all ages. This manuscript is a review about the use of antibiotics in neonates. Although it describes the use of antibiotics studied in different original studies, it is not made a really discussion about the rational use in neonates. They should not limit the presentation of studies on the most used antibiotics, they should discuss these data with the European and International guidelines. To support the title of the manuscript “Rational use of antibiotics in neonates”, authors should improve the critical discussion about this issue taking in account data from published studies that assess the inappropriate antibiotic use in neonates.

In page 1 – lines 35-37, the authors write that “consumption of antibiotics is commonly used as an indicator of rational medicine use”. Please clarify and justify these sentence.

In page 1 – lines 74-75 – Reference [9] should be replaced to the end of the sentence “In this dataset …. all antibiotics prescribed.”

Table 1, is not a useful information for the manuscript. Authors could move it to supplementary material.

Page 5 – line 169 – “… protein bound penicillins, like cefazolin or ceftriaxone …”. Please check this sentence, cefazolin and ceftriaxone are cephalosporines.

Figures 1 – is this figure constructed by the authors or it belongs to another published study? If yes please indicate the reference in the manuscript. Do the authors have permission to use it? The figure is for any drug administered by prolonged infusion? Or just for beta-lactam antibiotics? Please check the relevance to use this figure in the manuscript, and it location in the text.

Page 7 – line 267 (in the end) – “…that this in not…” change to “…that this is not…”

Author Response

The rational use of antibiotics is an important public health issue that needs to be discussed for all ages. This manuscript is a review about the use of antibiotics in neonates. Although it describes the use of antibiotics studied in different original studies, it is not made a really discussion about the rational use in neonates. They should not limit the presentation of studies on the most used antibiotics, they should discuss these data with the European and International guidelines. To support the title of the manuscript “Rational use of antibiotics in neonates”, authors should improve the critical discussion about this issue taking in account data from published studies that assess the inappropriate antibiotic use in neonates. We have considered the comment of the reviewer, and we obviously value his/her input. However, the major problem to assess rational or irrational use of drugs is that – in neonates - we still don’t have the tools (indication, dose, duration) to really quantify this, most relevant for the issue on ‘when’ to dose. This is already reflected in the abstract, and we have further added the need for tools in the second alinea. Moreover, we have further extended the subsection on when to prescribe with an alinea on biomarkers, and how guidelines can affect practices, but not necessary in the direction of reduced antibiotic use (NICE guideline, NeoPins study) in neonates [Mukherjee et al, Arch Dis Child Fetal Neonatal Ed 2004; Stocker et al, Lancet 2017]. Moreover, we have added some additional survey studies that illustrate the extensive variability in dosing regimens [Leroux et al, 2015; Metsvaht et al, 2015] Finally, we have also decided to adapt the title, to further reflect this aspect rather specific to neonates. In page 1 – lines 35-37, the authors write that “consumption of antibiotics is commonly used as an indicator of rational medicine use”. Please clarify and justify these sentence. This is explicitly mentioned in the reference 1 (website WHO), so we have added some lines as retrieved in reference 1 (Commonly used indicators of potential irrational practices are the incidence of polypharmacy, inappropriate self-medication, failure to adhere to clinical guidelines or dosing regimens, or inappropriate use of antibiotics (dose, indication, duration, route of administration) [1]. [1].). The specific papers are further discussed in the same alinea, and the text has been adapted to further highlight this In page 1 – lines 74-75 – Reference [9] should be replaced to the end of the sentence “In this dataset …. all antibiotics prescribed.” This has been adapted, thank you for this very specific suggestion to further improve the paper. Table 1, is not a useful information for the manuscript. Authors could move it to supplementary material. To the very best of our knowledge, this is an online only journal, so there is not much add on value to place the table either in the full paper or in the supplement. Since this provides additional information on the type and extent of antibiotics used in neonates, we still have the opinion that there is value in the table, especially for clinical researchers not routinely involved in neonatal clinical care. We hereby ask the editor on his/her opinion, and remain available to adapt, if this is perceived to be more valuable. However, based on the above mentioned rationale, we propose to keep this table in the revised version of the text. Page 5 – line 169 – “… protein bound penicillins, like cefazolin or ceftriaxone …”. Please check this sentence, cefazolin and ceftriaxone are cephalosporines. Sorry for this error, in an attempt to remain in line with the heading of the subsection, we suggest to adjust this to beta-lactams. Figures 1 – is this figure constructed by the authors or it belongs to another published study? If yes please indicate the reference in the manuscript. Do the authors have permission to use it? The figure is for any drug administered by prolonged infusion? Or just for beta-lactam antibiotics? Please check the relevance to use this figure in the manuscript, and it location in the text. The figure specifically focusses on beta-lactams, so this has been further stressed in the revised version of the legend of the figure. In this way, it is best to keep this figure also in this current location. This figure has been constructed specific for this review, as was mentioned in the accompanying letter at submission, but we assume that the reviewer had no access to this information. Consequently, there are no 3rd party or other copyright holders involved. Page 7 – line 267 (in the end) – “…that this in not…” change to “…that this is not…” Corrected, sorry for the textual error

Reviewer 3 Report

Dear Editors,

Thank you for asking me to review the paper 'Rational use of antibiotics in neonates'. This is a comprehensive paper discussing a number of strategies that are utilized in the prescribing of antibiotics in the neonates.

The authors describe the manuscript as a review paper. If this is a systematic review the review strategy should be included. If not, the authors should clarify that this is a narrative review.

The manuscript would be strengthened if the authors could briefly summarize the strengths and weaknesses of the strategies that they have described as well as include recommendations of how these strategies could be implemented at both a local and national level. In addition, it would broaden the scope of the paper if there was some comparison of how the outcomes of these strategies in the adult setting.

Author Response

Second reviewer: Thank you for asking me to review the paper 'Rational use of antibiotics in neonates'. This is a comprehensive paper discussing a number of strategies that are utilized in the prescribing of antibiotics in the neonates. We thank the reviewer for the overall very supporting and positive assessment of the paper and its topic. The authors describe the manuscript as a review paper. If this is a systematic review the review strategy should be included. If not, the authors should clarify that this is a narrative review. We agree, it is in our opinion obvious that this is indeed not a systematic review on all the aspects mentioned in the paper and we assume that this is also the opinion of this reviewer. For reason of clarity, we have added that this is indeed a narrative review on topics related to rational use of antibiotics in neonates. The manuscript would be strengthened if the authors could briefly summarize the strengths and weaknesses of the strategies that they have described as well as include recommendations of how these strategies could be implemented at both a local and national level. In addition, it would broaden the scope of the paper if there was some comparison of how the outcomes of these strategies in the adult setting. We thank the reviewer for this advice, and we have further extended on some ‘observations’ on the potentials and the limitations of biomarkers (NICE guideline vs NeoPins study), but also on the potential use of online drug formularies to further harmonize and improve drug prescription with the Dutch kinderformularium (van der Zanden et al, Arch Dis Child 2017) as an illustration of this approach. However, and because the paper mainly focused on the limitations of the currently available method tools, we prefer not to embark on a comparison on strategies in adults to neonates, since this feels – although a potential valuable effort – out of scope when we consider the topical aspects discussed in this narrative review.

Round  2

Reviewer 1 Report

The paper has been significantly and is now acceptable for publication.  Not all expected changes were made, but the overall writing is all right. 

Reviewer 2 Report

The manuscript has been improved and the authors have answer and clarified to all the comments.

A minor correction that was not understand by the authors:  In lines 81-55 – Reference [9] should be moved to the end of the sentence: “In this dataset …. all antibiotics prescribed [9].”

Reviewer 3 Report

The authors have addressed all the comments